# Further Mining and Characterization of miRNA Resource in Chinese Fir (*Cunninghamia lanceolata*)

**DOI:** 10.3390/genes13112137

**Published:** 2022-11-17

**Authors:** Houyin Deng, Rong Huang, Dehuo Hu, Runhui Wang, Ruping Wei, Su Yan, Guandi Wu, Yuhan Sun, Yun Li, Huiquan Zheng

**Affiliations:** 1Guangdong Provincial Key Laboratory of Silviculture, Protection and Utilization, Guangdong Academy of Forestry, Guangzhou 510520, China; 2National Engineering Research Center of Tree Breeding and Ecological Restoration, College of Biological Sciences and Technology, Beijing Forestry University, Beijing 100083, China

**Keywords:** Chinese fir, miRNA resource, bioinformatics, free energy, evolution

## Abstract

In this study, we aimed to expand the current miRNA data bank of Chinese fir (*Cunninghamia lanceolata* (Lamb.) Hook.) regarding its potential value for further genetic and genomic use in this species. High-throughput small RNA sequencing successfully captured 140 miRNAs from a Chinese fir selfing family harboring vigor and depressed progeny. Strikingly, 75.7% (n = 106) of these miRNAs have not been documented previously, and most (n = 105) of them belong to the novel set with 6858 putative target genes. The new datasets were then integrated with the previous information to gain insight into miRNA genetic architecture in Chinese fir. Collectively, a relatively high proportion (62%, n = 110) of novel miRNAs were found. Furthermore, we identified one MIR536 family that has not been previously documented in this species and four overlapped miRNA families (MIR159, MIR164, MIR171_1, and MIR396) from new datasets. Regarding the stability, we calculated the secondary structure free energy and found a relatively low R^2^ value (R^2^ < 0.22) between low minimal folding free energy (MFE) of pre-miRNAs and MFE of its corresponding mature miRNAs in most datasets. When in view of the conservation aspect, the phylogenetic trees showed that MIR536 and MIR159 sequences were highly conserved in gymnosperms.

## 1. Introduction

Plant mature miRNAs are generated from stem-loop precursors by a Dicerlike (DCL) enzyme and range from 20 to 24 nucleotides (nt) in length [1]. Their precursor miRNA sequence can fold into a stable stem-loop structure with lower minimal folding free energy than other sRNAs [2]. Additionally, mature miRNAs typically combine with Argonaute proteins in order to form RNA-induced silencing complexes (RISCs) with stronger stable structure, which can silence the expression of complementary mRNAs [3]. These properties of miRNAs ensure the regular performance of the growth and development of the organism. Plant growth and development are depended on gene expression and regulation [4]. MicroRNAs (miRNAs) are versatile regulators of gene expression and play a pivotal role in the evolutionary process of plants [5,6]. Conifers dominate the world’s forest ecosystems and are the most widely cultivated tree species [7]. The conifers are expected to provide important information regarding the evolution of highly conserved small regulatory RNAs [8]. An increasing number of miRNAs studies about growth and development are being carried out in coniferous trees from different genus, such as *Pinus* L. [8,9,10,11], *Picea* A. Dietrich [12], *Araucaria* Juss. [13] and *Ginkgo* L. [14]. To our knowledge, some miRNAs studies have also been conducted on several Cupressaceae trees, such as *Platycladus orientalis* (L.) Franco and Chinese fir (*C. lanceolata*) [15,16,17]. 

Chinese fir, an evergreen coniferous tree that occupies approximately 25% of plantations in southern China, has great commercial and ecological importance [18]. The different clones within this species harbor a broad range of growth variation, which means that a plethora of valuable genetic resources remain hidden in this species [19]. With the widespread accessibility of next-generation sequencing technology, small RNA sequencing and identification were performed for Chinese fir [20]. The miRNA expression profiles of this species were primarily conducted in seedling (stem and leaf), seed dormancy, vascular cambium activity, and vascular cambium development [16,20,21,22]. However, there is still a considerable lack of corresponding research on selfed progeny growth and development associated with miRNAs in Chinese fir, as selfing often results in inbreeding depression, marked by reduced survival and fertility, stunted growth and poor stress tolerafnce and even death [23,24]. Given this, the research on the field of selfing related to plant growth and development has received extensive attention [25,26,27]. 

Chinese fir is generally regarded as an outcrossing species with an effective mechanism to encourage outcrossing and prevent selfing [28]. Previous studies have shown that Chinese fir selfing often leads to evident inbreeding depression, which is manifested in selfed progeny by lower seed yield, reduced germination rate, stunted seedling growth and even death [29,30,31]. However, our previous study showed that a special Chinese fir clone, name ‘cx569’, produced a selfed progeny population with lower inbreeding depression in the seedling stage [31]. The seedlings presented two distinct phenotypes, showing higher vigor and lower vigor in terms of seedling height, number of roots, and fresh weight, etc. Therefore, the selfed progeny of the special material ‘cx569’ could provide additional genetic basis to fill the gaps on miRNA related to selfing in Chinese fir. Given this, we used a selfed family from this special material for miRNA mining and aimed to expand the Chinese fir miRNA resource through small RNA sequencing. Furthermore, we integrated our datasets with the available Chinese fir miRNA information to profile their genetic architecture. 

## 2. Materials and Methods

### 2.1. Plant Materials

The self-pollination of clone cx569 with the characteristic of self-fertility was conducted in a 2.5th generation seed orchard of Chinese fir in the Xiaokeng State Forest Farm (Dongguan, Guangdong Province, China; N24°70′, E113°81′, at an altitude of 328–339 m). The methods of pollination were adopted from a previous study [31]. The fresh seeds were collected from the self-pollination cones when they reached maturity. For collecting germinated seeds, the fresh seeds were surface sterilized and placed on seed germination boxes covered with gauzes moistened with sterile water, and the seed germination boxes were placed in a climatic incubator at 25 °C under a 24 h dark photoperiod with a humidity of 75%. Subsequently, the germinated seeds were then transferred to peat soil mixed with 20% expanded-perlite and maintained in a growth chamber at 25 °C under a 16/8 h light/dark photoperiod with a humidity of 75%. At 5 months old, the growth vigor of these progeny populations varied greatly with extremely depressed seedlings and normal seedlings (Appendix A). The normal seedlings were characterized by green stems and leaves, no visible pests or diseases and vigorous growth. Meanwhile, the depressed seedlings were shorter, dwarf, weak, chlorotic, reddish, leathery and subsequently withered compared to normal seedlings. To investigate whether there were differences in genetic basis between depressed seedlings and normal seedlings populations, we divided selfed progeny of cx569 into 2 groups according to seedling height. To reduce the error in the selection of depressed and normal seedlings, seven depressed seedlings (named ‘S21L’, ‘S38L’, ‘S42L’, ‘S189L’, ‘S218L’, ‘S220L’, and ‘S221L’) from the seedlings below the average seedling height and ten normal seedlings (named ‘S29L’, ‘S64L’, ‘S98L’, ‘S116L’, ‘S134L’, ‘S157L’, ‘S193L’, ‘S217L’, ‘S231L’, and ‘S256L’) from seedlings above the average seedling height were selected by using the random function of Microsoft Excel. 

### 2.2. RNA Extraction, RNA Library Construction and Functional Annotation

Total RNA was extracted from the above-ground parts (stem and leaf) of seven depressed seedlings and ten normal seedlings using the RNeasy Plant Mini Kit (Qiagen, Hilden, Germany). The RNA concentration and integrity were evaluated using a Nanodrop 2000 spectrophotometer (Thermo Fisher Scientific, Wilmington, DE, USA) and an Agilent 2100 Bioanalyzer (Agilent Technologies, Santa Clara, CA, USA), respectively. 

In order to obtain raw data, library construction and deep sequencing were conducted on an Illumina system HiSeq XTen platform provided by Biomarker Biotechnologies Corporation (Beijing, China, www.biocloud.net (accessed on 15 January 2022)). These raw data (17 libraries) were uploaded in the Sequence Read Archive (SRA) database of the NCBI with accession number PRJNA899478. After removal of low-quality reads, short reads, and adapters, the clean reads were generated for de novo assembly using Trinity [32]. Sequence alignment was performed using Bowtie software [33]. The mapped reads consisting of 48,755 unigenes were used for further analysis. The unigene function was annotated based on the following six databases: NCBI nonredundant protein sequences (nr), Kyoto Encyclopedia of Genes and Genomes (KEGG) and Gene Ontology (GO), Swiss-Prot, Protein family (Pfam) and Clusters of Orthologous Groups of proteins (KOG/COG). Additionally, the obtained Chinese fir transcriptome database was used for reference sequences in pre-miRNA prediction and identification of potential miRNA targets.

### 2.3. Identification of miRNAs and Prediction of Their Target Gene

The clean reads were mapped to various RNA databases including the Silva, GtRNAdb, Rfam, and Repbase databases for filtering ribosomal RNA (rRNA), transfer RNA (tRNA), small nuclear RNA (snRNA), small nucleolar RNA (snoRNA) and repeats using Bowtie. The remain reads were aligned to the Chinese fir mRNA transcriptome by Bowtie. To identify conserved miRNA, the drive reads were aligned with currently mature miRNA sequences from all plant species in the miRbase Sequence Database (https://www.mirbase.org/, (Release 22.1) accessed on 1 August 2022) using a BLASTn search with up to one mismatch [34]. Sequences with identity to other species were identified as known miRNA. The unannotated reads were processed for novel miRNA identification by miRDeep2 with adjusted parameters [35]. To further reveal known and novel miRNA precursors, the mapped reads were used to extract potential precursors using miRDeep2 with adjusted parameters [35]. The precursors were aligned to short reads, and statistical analyses of read distributions were generated for each miRNA gene for evaluation of miRNA. To quantify the abundance of miRNA, the transcripts per million (TPM) value was defined as ‘counts of reads mapped to miRNA × 1,000,000 reads mapped to the reference genome’. The prediction of target genes of the mature miRNA from the known and novel precursor miRNAs (pre-miRNAs) was performed by TargetFinder with default parameters [36].

### 2.4. Bioinformatics Analysis

In order to integrate our datasets with the available Chinese fir miRNA information to profile the genetic architecture, we collected all Chinese fir miRNAs from miRbase and published small-RNA sequencing datasets for analysis. For the minimal folding free energies (MFEs) of secondary structures of miRNAs analysis, we derived all precursor sequences and their corresponding mature sequences from Chinese fir miRNA (previously reports included) and subjected them to Mfold web server (http://www.unafold.org/ (accessed on 10 August 2022)) with default parameters [37]. 

Venn analysis was conducted to acquire unique miRNAs and overlapped miRNAs. For the phylogenetic analysis, the precursor miRNA sequences from Chinese fir and other plant species were subjected to MAFFT software (version 7) for multiple sequence alignment [38]. The phylogenetic tree was constructed on the PhyloSuite platform (version 1.2.2) using the maximum likelihood method with 1000 bootstrap replicates under the optimal model selected by IQ-TREE [39,40]. All phylogenetic tree modifications were completed using the ITOL online tool (https://itol.embl.de/ (accessed on 15 August 2022)) [41]. 

## 3. Results

### 3.1. An Overview of High-Throughput Sequencing Datasets

A total of 290,752,309 raw reads were obtained from 17 small RNA libraries (Appendix A). After we filtered out low-quality reads and reads <18 bp or >30 bp in length and removed adaptor sequences and reads with unknown base N content ≥10%, the clean reads accounted for 99.94% (290,566,075) of the total raw reads, and the Q30 values for all samples were greater than 95.31%, indicating the reliability of the data. In the seventeen clean data libraries, the length of the sRNAs (18–30 nt) presented a different distribution pattern, with a major peak at 21 nt, followed by 20 nt or 22 nt sRNAs (Appendix A). This differs from earlier findings in Chinese fir and other conifer species, such as *Araucaria angustifolia*, in which the length of sRNAs had a highest abundance at 21 nt and a minor peak at 24 nt [13,20,21]. The result suggests that the sRNAs datasets obtained from the special materials used in this study are diverse compared to the datasets of sRNAs previously recorded in Chinese fir. Furthermore, the sRNA size distribution in most of conifer species differed from those of many angiosperms that are mainly clustered in 24 nt [8,42,43,44]. This implied that there was an unusual RNA silencing signature in gymnosperms. One reason for this may be the presence of a specific DCL3 enzyme, a member of the Dicer-like family (DCL) gene that cleaves double-strand RNAs, matures 24-nt RNAs in angiosperms [45]. The clean read sequences were then compared with the Silva, GtRNAdb, Rfam, and Repbase databases to further obtain rRNAs, tRNAs, snRNAs, snoRNAs, and repeat-associated sRNAs, and the unannotated reads containing small regulatory RNAs were used to identify miRNAs. More than 11 million sRNAs were obtained from each library (Figure 1; Appendix A). Among them, unannotated reads were the most abundant RNAs with at least 7 million read counts. The large proportions of unannotated reads in this study agree with previous findings in Chinese fir and other gymnosperms such as *Cryptomeria fortunei* [21,46]. However, fewer unannotated sRNAs are recorded in many angiosperms [47]. Therefore, there may be abundant miRNA resources within coniferous species. The rRNAs were the second largest group with a range from 3,820,953 to 9,114,419 read counts. Other sRNAs possessed relatively low reads. In the next step, unannotated reads were subsequently mapped to the Chinese fir mRNA transcriptome database. Approximately 31.41–37.43% of the sRNAs were successfully matched to the reference transcriptome (Appendix A). The mapped reads were further used to detect miRNAs.

### 3.2. Classification of the miRNAs

It is well known that miRNAs can be divided into known and novel miRNAs. Known miRNAs are based on various databases matched to highly homologous sequences, while novel miRNAs are predicted by computerized algorithms under strict criteria. miRbase Database (Release 22.1) is the largest miRNA resource bank, which contains 38,589 miRNAs to date (1 August 2022). To identify known miRNA, mapped reads were aligned with currently known mature miRNA sequences from all plant species in the miRbase Database using a BLASTn search with up to one mismatch. Sequences that were highly homologous to other species were identified as known miRNAs. As a result, 30 sequences successfully matched 16 conserved families (Figure 2). Within these miRNA families, the MIR159 and MIR396 families were the largest with 5 members, which comprised multiple species of different mature sequences. The second group were MIR319, MIR398, MIR399 and MIR529 family, consisting of 2–4 members. The remaining 10 families, including MIR156, MIR160, MIR162_1, MIR164, MIR166, MIR171_1, MIR395, MIR408, MIR536 and MIR6725, contained only one miRNA. For the length of miRNAs, the majority of the lengths of known mature miRNAs were 21 nt, and most of the lengths of their corresponding precursor sequences were 200–250 nt (Appendix A). The prediction of target genes for miRNAs is a key step in understanding the regulation of miRNAs. In this study, we found that the identified known miRNAs harbored 1148 target genes with various functions by prediction and annotation (Appendix A). 

A typical stem-loop precursor is a prerequisite for the identification of newly developed miRNAs according to the criteria for the annotation of novel miRNAs [1]. For mapped reads that were not matched to known miRNAs, their sequences were processed for novel miRNA identification. In our study, 110 novel miRNAs and their precursor sequences were positively predicted with a computerized algorithm using miRDeep2 software with adjusted parameters [35]. These Chinese fir novel miRNAs were given names in the form of ‘cln-miR plus number’, e.g., cln-miR01 (Appendix A). Similar to identified known miRNAs, 59% (n = 65) of the number of novel miRNA lengths primarily clustered at 21 nt, and the precursor length had a peak at 200–250 nt (Figure 2B,C). Additionally, the novel miRNAs targeted to 6927 hypothesis genes with extensive function indicating that these novel miRNAs may act as a pivotal genetic regulator during selfed seedling growth and development (Appendix A). For example, some MYB transcription factors are involved in plant-form development of selfed progeny through associating with lignin biosynthesis-related genes [48]. Moreover, many novel miRNAs targeted to MYB transcription factor genes in this study, so there may be a similar mechanism here that regulates the plant-form development of selfed progeny. To compare global expression profiles of detected miRNAs during seedling growth and normal development and depression, the expression analysis of miRNAs was performed with two different types of seedlings based on the normalized read count (transcripts per million, TPM) for known and novel miRNAs. In this study, we found that known and novel miRNAs did not show divergent expression patterns between normal and depressed seedlings (Appendix A), but presented different expression patterns between some individuals, indicating that these individuals may harbor differences in their regulatory processes in response to adaptive growth. 

### 3.3. The Newly Developed miRNAs

Knowledge on available miRNA resources within a species is important for revealing the miRNA atlas in this species and will be very useful for further genetic and genomic use in this species. In light of this, we further integrated our datasets with the available Chinese fir miRNA information to profile the genetic architecture in this species, which mainly involves different biological processes, such as vascular cambium activity, vascular cambium development and seed dormancy, etc. [20,21,22]. A total of 899 known miRNAs were found in current reports based on statistical analysis (Figure 3A; Appendix A). A larger proportion (72%, n = 651) of the known miRNAs were found by Liu et al. [20] compared to other Chinese fir studies. In contrast, Cao et al. [19] detected a smaller set of known miRNAs (n = 19). In this study, we also identified a relatively small set of known miRNAs (n = 30) in comparison with other records. Concerning the miRNA family, Venn analysis revealed four overlapped MIRNA families (MIR159, MIR164, MIR171_1 and MIR396) in all known Chinese fir miRNAs (Figure 3B). There were also a series of miRNA families in different overlapped combinations. Strikingly, one miRNA family belonging to MIR536 detected in this study had not been previously documented in Chinese fir. In addition, a high ratio (62%, n = 110) of novel miRNA numbers was found in our study when compared to all integrated novel miRNAs (n = 178) of Chinese fir (Figure 3C; Appendix A). These results showed that the new set of miRNAs identified in this study was very different from other documents in Chinese fir. Similar to our findings, in *C. fortunei*, Hu et al. [49] revealed 80 conserved miRNAs belonging to 32 families and 145 unvalidated new miRNAs based on different developmental stages of the vascular cambium, whereas Zhang et al. [46] obtained more miRNAs, including 517 known and 212 novel miRNA mature/star sequences under the background of needle discoloration. Complicated biological processes of different developmental tissues within same species may contribute to the type and quantity of miRNAs. These results suggest that the further mining of miRNAs in this study is meaningful for clarifying the genetic architecture in this species. 

For the sequence length of the known and novel mature miRNAs, the miRNA in every study also had a peak at 21 nt (Figure 3D,E). Meanwhile, 585 known pre-miRNAs with lengths ranging from 20 to 415 nt were detected in Liu et al. [20], followed by this study harboring 30 known pre-miRNAs (83–250 nt); other studies harbored fewer known pre-miRNAs, ranging from 4 to 19 known pre-miRNAs (Appendix A). Interestingly, the largest number (n = 110) of novel pre-miRNAs was successfully predicted in this study in comparison with the intergraded data (n = 178). miRNAs regulate target gene expression post-transcriptionally through sequence complementarity, which is involved in many developmental processes in plants [50]. Although fewer known miRNAs (n = 30) were identified in this study, the number of target genes (n = 1148) was relatively high in our data when compared to that of target genes detected by Wan et al. [38] (n = 209), Qiu et al. [21] (n = 240), Cao et al. [22] and Liu et al. [20] (n = 4077) (Appendix A). Additionally, we found a large number (n = 6927) of potential putative target genes in this study, while others recorded up to 660 target genes. These results indicated that the miRNAs detected in this study may play important roles in plant growth and development in the background of self-fertilization. 

In order to investigate the number of newly developed miRNAs in this study, all integrated known and novel mature miRNA sequences were blasted by ClustalX2 software, respectively. As a result, 19 known and 5 novel mature miRNA sequences in this study were found to be identical to those previously documented (Appendix A). These results showed that 11 known and 105 novel miRNAs with 6858 target genes in this study were newly expanded miRNAs sets for Chinese fir (Appendix A). The detected overlapped miRNAs and unique miRNAs in this study may help to understand the genetic variation in trees.

### 3.4. An Integrated Analysis of the Stability of Chinese Fir miRNAs in Terms of Energy 

Precursor miRNA sequences do not have tight interactions with proteins before their cleavage by Dicer (or Dicer-like) enzymes [51]. Therefore, a stable secondary structure with low minimal folding free energy (MFE) might be necessary for miRNAs to avoid early degradation. Regarding the stability, we calculated the MFE for both the pre-miRNA and mature miRNA with Mfold web server under the default parameters [37]. Only the results of the optimal folding with minimum MFE value were used (Appendix A). In this study, the MFE of known pre-miRNAs ranged from −39.75 to −97.4 kcal/mol with an average negative folding value of −72.4 kcal/mol (Appendix A). In contrast, the absolute MFE value of their mature miRNAs was less than 5. For the MFE of novel miRNAs, their MFE of precursor sequence spanned from −11.1 to −115.50 kcal/mol with an average negative folding value of −67.6 kcal/mol. The absolute MFE value of the corresponding mature miRNAs was still less than 5. Regression analysis further revealed a relatively low R^2^ value (R^2^ < 0.22) between the MFE of pre-miRNAs and the MFE of the corresponding mature miRNAs in most studies (Figure 4). This result suggests that the thermodynamic stability of the secondary structure does not seem to be of major importance for mature miRNAs. One of the important reasons for this is that mature miRNAs typically combine with Argonaute proteins in order to form RNA-induced silencing complexes (RISCs) with stronger stable structure, which can silence the expression of complementary mRNAs [3].

### 3.5. Phylogenetic Analysis of the Conserved miRNAs between Chinese Fir and Other Plant Species

Based on the one unique miRNA family (MIR536) and the four overlapped (conserved) miRNA families (MIR159, MIR164, MIR171_1, and MIR396) in Chinese fir miRNA datasets in this study, we attempted to elucidate the evolutionary history of these conserved family miRNAs among plant species. Phylogenetic analysis of precursor sequences contributes to the understanding of the evolutionary relationships of individual MIRNA genes in a family [52]. For the MIR536 family members in most plant species, we selected all sequences belonging to MIR536 family member from the current miRbase database, which comprised one monocotyledon (*Asparagus officinalis* L.), two eudicotyledons (*Populus trichocarpa* Torr. & Gray and *Citrus sinensis* (L.) Osbeck), one gymnosperm (*picea abies* (L.) Karst.), one moss (*Physcomitrella patens* Bruch & Schimp.) and one lycophyte (*Selaginella moellendorffii* Hieron.). Meanwhile, we searched for all pre-miRNA sequences of MIR159, MIR164, MIR171_1 and MIR396 families in 11 land plant species, including *Oryza sativa* L., *Zea mays* L., *Populus trichocarpa*, *Arabidopsis thaliana* (L.) Heynh., *Brassica napus* L., *Medicago truncatula* Gaertn., *Glycine max* (Linn.) Merr., *Pinus taeda* L., *Pinus densata* Mast. and *Picea abies* from the current miRbase database (Release 22.1) and also included the sequences of *Platycladus orientalis* (L.) Franco, a close relative of Chinses fir [14]. 

In our analysis of the phylogenetic relationships of the MIR536 family, it is clear that the Chinese fir MIR536 clustered with other gymnosperms, indicating its high evolutionary conservation (Figure 5). Despite a lack of empirical evidence regarding the presence of cln-MIR536, it is undeniable that the MIR536 families are predominant in gymnosperms [53]. Most of the MIR159 gathered with other gymnosperms, suggesting that Chinese fir MIR159 is also deeply conserved in gymnosperms during evolution (Appendix A). This is consistent with early findings that miR159 families in gymnosperms formed a separate distinct clade, suggesting a highly conserved relationship of these families in plant species [54]. In contrast, some MIR164 and MIR396 precursor sequences clustered in monocotyledon and eudicotyledons species, suggesting their diversification of evolutionary divergence (Appendix A). In the phylogenetic tree of the MIR171_1 family precursor sequence (Appendix A), cln-MIR171 was distributed in all three pedigree lines including lycophytes, monocotyledon, and eudicotyledons, while the distribution of MIR171 in other gymnosperms was relatively scattered, implying the diversification of cln-miR171. However, Qiu et al. [55] revealed pde-MIR171 to be clustered with other gymnosperms, in which angiosperms form a separate cluster indicating a deep evolutionary conservation of the miR171 families. The reason for this may be that all miR171 family sequences of selected species were used for phylogenetic analysis in this study, while Qiu et al. [55] only used homologous sequences from different plant species. The result suggests that the different homologous sequences of MIR171_1 family have interspecific differences in evolutionary relationships.

## 4. Conclusions

In the present study, 140 miRNAs were obtained from a Chinese fir selfing family harboring vigor and depressed progeny with rather different phenotypes. Remarkably, 75.7% (n = 106) of these miRNAs have not been recorded previously, and most (n = 105) of them belong to the novel set with 6858 putative target genes. Based on published small-RNA sequencing datasets from Chinese fir, we found a high ratio (62%, n = 110) of novel miRNA numbers in our study when compared to all integrated novel miRNAs (n = 178) of Chinese fir. Furthermore, we identify one MIR536 family that had not been previously documented in this species and four overlapped miRNA families (MIR159, MIR164, MIR171_1 and MIR396) from Chinese fir miRNA datasets. Additionally, we also revealed a relatively low R^2^ value (R^2^ < 0.22) between the low minimal folding free energy (MFE) of pre-miRNAs and the MFE of its corresponding mature miRNAs in most datasets. From the conservation aspect, phylogenetic analysis shows that both MIR536 and MIR159 are conserved in gymnosperms, whereas the other miRNA families had evolutionary diversification. The present data provide potential value regarding Chinese fir miRNA biology and will be very useful for further genetic and genomic use in this species.

## Figures and Tables

**Figure 1 genes-13-02137-f001:**
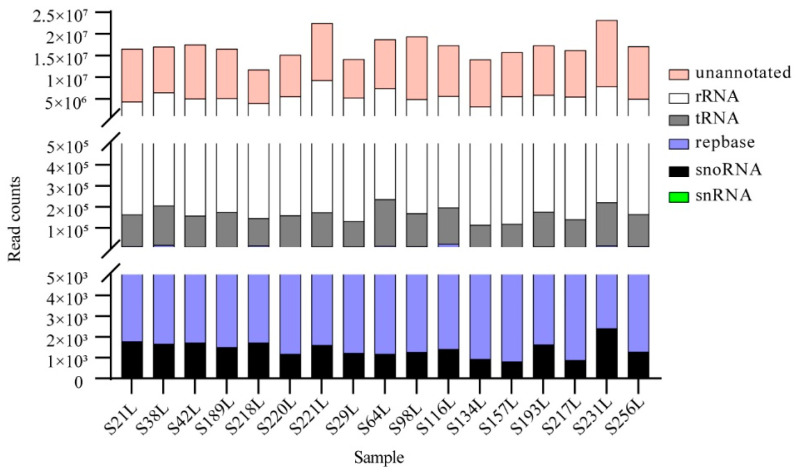
Sample read sequence types and attributes.

**Figure 2 genes-13-02137-f002:**
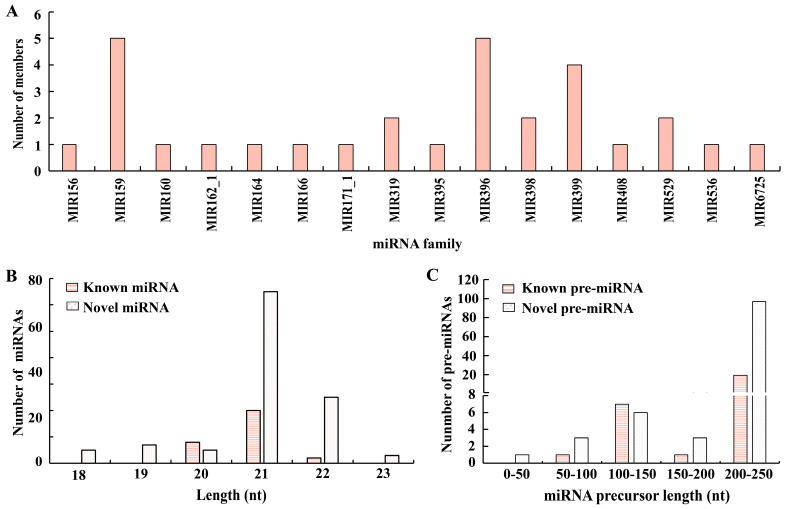
Characteristic of miRNA in Chinese fir. (**A**) Identification of miRNA family and its corresponding numbers from a selfed family in Chinese fir. (**B**) indicates the length distribution of known and novel miRNAs and its corresponding quantities. (**C**) indicates the length distribution of known and novel pre-miRNA and its corresponding quantities.

**Figure 3 genes-13-02137-f003:**
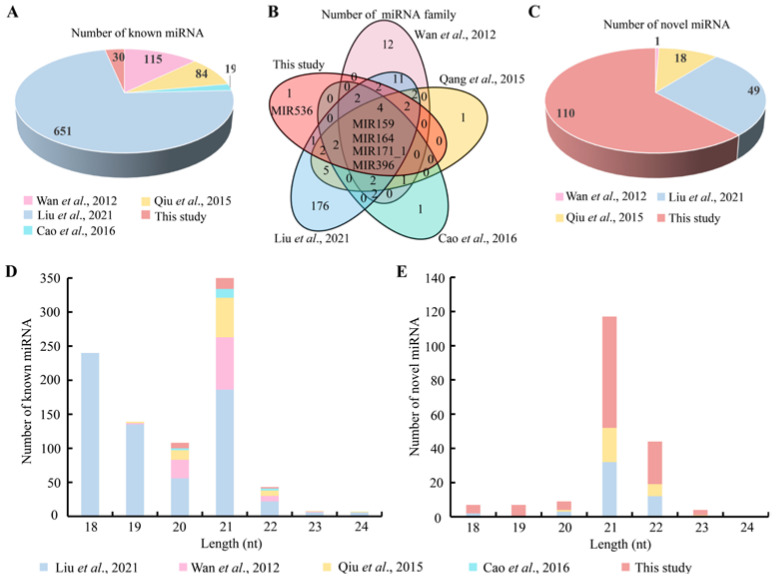
The number of miRNAs and the known miRNA families from published Chinese fir miRNA datasets. These Chinese fir miRNA datasets were collected from Wan et al., 2012 [16]; Qiu et al., 2015 [20]; Cao et al., 2016 [21]; Liu et al., 2021 [22]; and this study. (**A**) Pie chart showing the number of known miRNAs from different Chinese fir miRNA datasets. (**B**) Venn diagram presenting the number of miRNA families from different Chinese fir miRNA datasets and revealing four overlapped miRNA families (MIR 159, MIR164, MIR 171_1 and MIR 396) in all miRNA datasets and one unique miRNA family (MIR 536) in this study. (**C**) Pie chart showing the number of novel miRNAs from different Chinese fir miRNA datasets. (**D**) indicates the length distribution of known miRNAs and its corresponding quantities. (**E**) indicates the length distributions of novel miRNAs and the corresponding quantities.

**Figure 4 genes-13-02137-f004:**
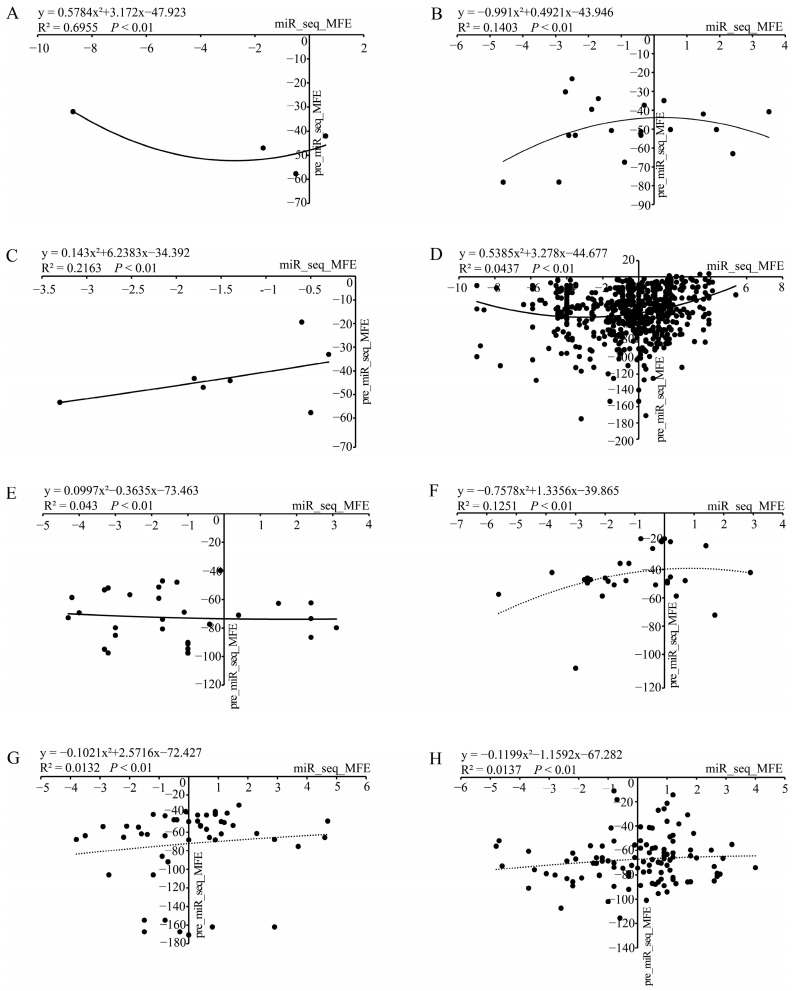
The relationship of minimal folding free energy (MFE) between pre-miRNA and the corresponding mature miRNA. Pre_miR_seq_MFE (precursor miRNA sequence MFE) and the corresponding miR_seq_MFE (mature miRNA sequence MFE) of known miRNAs in (**A**–**E**) from Wan et al., 2012 [16]; Cao et al., 2016 [20]; Qiu et al., 2015 [21]; Liu et al., 2021 [22]; and this study, respectively. Pre_miR_seq_MFE and the corresponding miR_seq_MFE of novel miRNAs in (**F**–**H**) from Qiu et al., 2015 [21]; Liu et al., 2021 [22]; and this study, respectively.

**Figure 5 genes-13-02137-f005:**
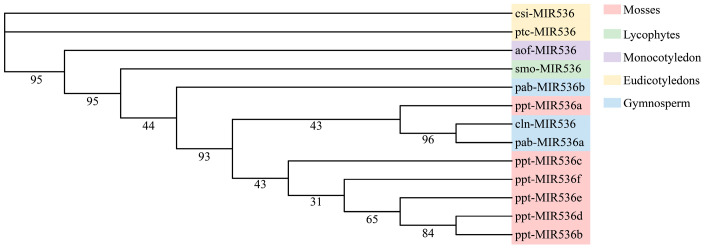
Phylogenetic tree of pre-miR536 in mosses, lycophytes, monocotyledons, eydicotyledons, and gymnosperms obtained from the IQ-TREE using the maximum likelihood method with 1000 bootstrap replicates. Labels: pab—*P*. *abies*; cln—*C. lanceolata*; smo—*Selaginella moellendorffii*; csi—*Citrus sinensis*; ppt—*Physcomitrella patens*; aof—*Asparagus officinalis*.

## Data Availability

The dataset(s) supporting the conclusions of this article are available in the NCBI’s Short Read Archive (SRA) with the BioProject accession numbers PRJNA899478.

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
