# Peer review of "Further Mining and Characterization of miRNA Resource in Chinese Fir (Cunninghamia lanceolata)"

_genes, 2022, doi:10.3390/genes13112137_

Round 1
Reviewer 1 Report
The presented article is a robust description of the presence of miRNAs in Cunninghamia lanceolata. I think it will be of interest to readers. I have only a few technical comments:
L 45 Names of authors of taxon shouldn’t be in italic
L51 & L54 “specie” should be changed to “species”
L52 & L58 Once is “Small RNA sequencing”, next is “small RNA Sequencing” – it should be one pattern of using the capital letter
Figure 3 Text under the plots is quite small, I think it should be larger
L266 “And” at the beginning of the sentence is unnecessary
L282 there is an additional dot at the end of the description, which should be removed
L296 “Picea abie” should be “Picea abies”; additionally, many species names appeared in paragraph 3.5 of the discussion for the first time – thus, it should be presented with the author name
Author Response
From:
Prof. Dr. Huiquan Zheng,
Guangdong Provincial Key Laboratory of Silviculture, Protection and Utilization; Guangdong Academy of Forestry, 510520 Guangzhou, People’s Republic of China
Tel: +86-20-87584306;
Fax: +86-20-87031245;
E-mail: zhenghq@sinogaf.cn
November 12, 2022
To:
Dear genes reviewer,
We are very glad to re-submit our manuscript (genes-1992666) to your journal, genes. The manuscript has been revised according to the thoughtful comments and suggestions from the reviewers and editor again. We are very sure that the revised manuscript is more readable. All co-authors have made a great contribution to this project and support our submission.

Reviewer 2 Report
The purpose of this article is to provide effective microRNA data to the public, so the sequencing data of microRNA should be deposited in a database like NCBI SRA...
The introduction part of the relevant research progress is insufficient. The author mentioned the reports on the microRNA research of Chinese fir with no elaborate, which cannot prove the authors statement that “the lack of corresponding research on Chinese fir”.
The reasons for the division of normaThe purpose of this article is to provide effective microRNA data to the publicl and depressed seedlings in this paper are puzzling. Why you chose to divided the populations into these two groups? There is no introduced in the part of Introduction, and the aim of the paper is not explained the purpose of the two groups selected. In addition, what kind of seedlings are growing normal or depressed? What are the criteria? Seedlings height, root length or biomass? What about the contrast photos of the morphological features? I don't see any relevant basis, please add!
Where is the part of Discussion?
Line 139: the length of the sRNAs (18-30 nt) shared a similar distribution pattern, with a major peak at 21 nt, followed by 24 nt sRNAs (Table S2).
It seems that the secondary peak is not 24 nt according to Table S2.
Line140: [13,18,20,] is charge to [13,18,20].
Line190: Figure 2 C and Figure 2 D is charge to Figure 2 B, Figure 2 C and Figure 2 D. And Figure 2B and Figure 2C can be made into a figure like Figure 2D.
Line190: Additionally, the novel miRNAs targeted to 6927 hypothesis genes with extensive function indicating that these novel miRNAs may play a pivotal genetic regulator during selfed seedling growth and development (Table S8).
Maybe more explanation is needed to prove that these novel miRNAs may play as a “pivotal genetic regulator” during selfed seedling growth and development.
Line196: In this study, we found that known and novel miRNAs showed divergent expression patterns between normal and depressed seedlings (Figure S1), indicating that their regulatory pro- cesses in response to adaptive growth are different.
It is difficult to conclude from the expression heatmap and clustering results in Figure S1 that there are different “expression patterns” between depressed and normal seedlings with different characteristics.
Line201, 245: Figure 2 and Figure 3 should add more information to annotate the names and meanings of each figure.
Line275: Figure 4 has a low resolution. It would be better to change the order of the small figures in Figure 4, unify the Spaces or underscores between the same type of images, and adjust the overlap in Figure 4H.
Supplemental tables should be made into a three line table.
Supplemental table2: Table S3 is changed to Table S2.
The format of references should be carefully revised according to the requirements of the journal. For example, For example, there is a punctuation error in line 355.

Author Response

(The authors gave the same response as above.)
